# Effects of Post Traumatic Growth on Successful Aging in Breast Cancer Survivors in South Korea: The Mediating Effect of Resilience and Intolerance of Uncertainty

**DOI:** 10.3390/healthcare11212843

**Published:** 2023-10-28

**Authors:** Su Jeong Yi, Ku Sang Kim, Seunghee Lee, Hyunjung Lee

**Affiliations:** 1College of Nursing, Dankook University, Cheonan 31116, Republic of Korea; 12181056@dankook.ac.kr; 2Department of Breast Surgery, Kosin University Gospel Hospital, Busan 49267, Republic of Korea; ideakims@gmail.com; 3College of Nursing, Chungnam National University, Daejeon 35015, Republic of Korea

**Keywords:** breast cancer, post traumatic growth, resilience, intolerance of uncertainty, nursing intervention

## Abstract

This study aimed to identify post-traumatic growth and successful aging and the mediating effects of resilience and intolerance of uncertainty in breast cancer survivors. This study employed a descriptive survey approach. Data were collected from 143 breast cancer survivors between 27 January and 10 December 2021, at a cancer center in Gyeongsangnam-do, South Korea. SPSS/WIN 25.0 and PROCESS Macro version 3.5 were used for data analysis. Descriptive statistics were analyzed with SPSS. PROCESS was used to conduct a mediation analysis and the significance of the mediating effect was evaluated using 95% confidence intervals. Successful aging was significantly associated with post-traumatic growth, resilience, and the intolerance of uncertainty. The impact of post-traumatic growth on successful aging was mediated by resilience in breast cancer survivors. Resilience should be considered when developing nursing interventions to enhance post-traumatic growth and promote successful aging in breast cancer survivors.

## 1. Introduction

The incidence of patients with breast cancer in South Korea increased at an average annual rate of 4.3% from 2007 to 2019. Breast cancer has a high survival rate, with a five-year relative survival rate of 93.6% after a diagnosis [1]. The average age of patients with breast cancer at diagnosis is 52.5 (±8.23) years; the highest incidence occurs in individuals in their 40 s, with a growing number of cases reported in those who are in their 50 s and beyond [2]. 

The increasing incidence of breast cancer and the extended average survival time after diagnosis means that there will be a growing number of breast cancer survivors who will live longer. With an the increase in the breast cancer survival rate, many patients are becoming more concerned about their long-term health, including the quality of life and successful aging after cancer. Therefore, there is a need for attention to successful aging in later life among cancer survivors who have returned to their daily lives [3]. Successful aging is an individual’s sense of having adapted to the physiological and functional changes associated with the passage of time, while also finding meaning or purpose in life [4]. Successful aging is a broad, complex concept that encompasses attributes of both physical and psychological health [5] and means having a low risk of disease and disability, maintaining high levels of physical and mental functioning, and actively participating in life [6]. Patients with breast cancer undergo aggressive treatment-including chemotherapy-depending on the stage of the cancer, even after surgery, owing to the risk of cancer metastasis and recurrence. After treatment ends, they could have many negative experiences, such as treatment side effects, complications, and changes in femininity [7]. As breast cancer survivors age, they simultaneously experience the functional and physiological challenges associated with the normal aging process, as well as the various challenges associated with treatment. Therefore, their aging process is believed to require more active management and care. Focusing on the strategies and processes of successful aging, that is, coping with life after treatment and providing appropriate interventions, is vital [8]. Flood’s theory [4] of successful aging, considers it an adaptation of functional performance, intrapsychic factors and spirituality; intrapsychic factors refer to an individual’s character that can enhance or impair their ability to adapt to change and solve problems. Therefore, successful aging requires an exploration of the individual’s character. Consequently, this study seeks to provide an integrated understanding of the psychological mechanisms that influence successful aging.

Post-traumatic growth is associated with adaptation and successful aging in patients with cancer [8]. Further, in a study of middle-aged women, post-traumatic growth was a positive predictor of successful aging. Post-traumatic growth is defined as psychological well-being resulting from the positive adjustment to trauma [9], and a previous study [10] has shown that psychological well-being is significantly and positively related to successful aging. Therefore, it is expected that there is a significant relationship between post-traumatic growth and successful aging. Post-traumatic growth is the positive change in perceptions of self, others, and life after a traumatic event and includes changes in self-perception, interpersonal relationships, and life stance [11]. As they enter remission after the end of medical treatment, many breast cancer survivors experience physical symptoms such as lymphedema, pain, and fatigue, as well as mental distress such as depression and fear of recurrence [12]. However, in addition to these negative aspects, breast cancer survivors experience positive changes such as an increased appreciation for life, setting life goals, and discovering their strengths [13,14]. These changes can be referred to as post-traumatic growth; from a human developmental perspective, effective strategies to promote and sustain post-traumatic growth are needed for successful transition and adaptation to older age as a breast cancer survivor [13]. This is because post-traumatic growth in patients with cancer is a major factor in improving their coping skills, enabling them to actively face challenges, and ultimately elevating their level of successful aging in the future. If post-traumatic growth influences successful aging, how is the relationship between the two variables established? If we introduce a mediator in the relationship between these independent and dependent variables, we can understand the “how” in the relationship between the two variables [15]. By utilizing a mediation model, we can examine the internal psychological resources an individual possesses to foster successful aging and understand its mechanism. 

We selected resilience as the first parameter. A key concept that could reduce negative outcomes through positive coping is resilience in breast cancer survivors [16]. Resilience is defined as the ability to return to an original state after being modified by an external force [16]. It refers to the ability to psychologically overcome adversity, leading to positive outcomes or reducing negative outcomes [17]. Post-traumatic growth and resilience are correlated [18,19]. An important factor in successful aging, despite the various changes throughout the life cycle, is individuals’ resilience [20]. Jeste et al. [21] reported that higher levels of resilience are associated with higher successful aging scores in the elderly; thus, it is worthwhile to explore the mediating effect of resilience on the relationship between post-traumatic growth and successful aging in breast cancer survivors. In addition, previous studies have indicated that there is either a negative correlation between resilience and the intolerance of uncertainty [22], or resilience on intolerance of uncertainty [23]. This suggests a need for further exploration of their relationship.

The study also highlighted intolerance of uncertainty as another variable. Since cancer progression and prognosis cannot be predicted with certainty, cancer survivors must endure uncertainty, both in general and in specific aspects related to the disease [24]. Particularly for breast cancer, in which survival rates are high, intolerance of uncertainty can lead to negative reactions across cognitive, emotional, and behavioral domains to stimuli or situations that arise in everyday life [25]. People with intolerance of uncertainty perceive and respond negatively to uncertain information and situations, regardless of probability or outcome [26], and they demonstrate increased psychological distress when faced with ambiguous symptoms [27]. Previous studies revealed that intolerance of uncertainty about disease prognosis was also associated with depressive symptoms [28,29], health anxiety, and anxiety sensitivity in female patients with breast cancer [30]. However, most previous research has focused on understanding the association between disease-related uncertainty and psychological outcomes [28,29,30]. In addition, the scant literature has explored the impact of intolerance of uncertainty as a factor related to cognitive, emotional, and behavioral responses to post-traumatic growth and successful aging in breast cancer survivors.

Antonovsky’s [31,32] Generalized Resistance Resource (GRRs) theory primarily addresses how individuals maintain health and adapt when confronted with stress or challenges. Antonovsky [31,32] suggested that GRRs are resources or abilities that assist individuals in coping with negative situations such as stress or illness. Post-traumatic growth can be viewed as an indicator of the GRRs that breast cancer survivors acquire after experiencing trauma. Resilience can be considered a GRR, and intolerance of uncertainty can be considered an indicator of a lack of or insufficient GRRs. Post-traumatic growth will enhance resilience and reduce intolerance of uncertainty. This enhanced resilience and reduced intolerance of uncertainty will promote successful aging. 

In the existing research, there is a dearth of studies examining the relationship between post-traumatic growth and successful aging among breast cancer survivors, especially the mediating effect of intolerance of uncertainty and resilience. To address this research gap, this study explores the relationship between post-traumatic growth and successful aging among breast cancer survivors from a new perspective and reveals the important role of intolerance of uncertainty and resilience. The specific aims were (a) to examine correlations among variables, including post-traumatic growth, successful aging, intolerance of uncertainty, and resilience; and (b) to examine the serial multiple mediation of resilience, and intolerance of uncertainty between post-traumatic growth and successful aging. The serial multiple mediator model has the advantage of providing a more comprehensive understanding of psychological mechanisms by examining the previously individually studied variables within a model that includes two mediating variables [15]. It is expected that this will provide insights into the process of improving quality of life and successful aging among breast cancer survivors.

## 2. Materials and Methods

### 2.1. Study Design

This was a descriptive correlational study.

### 2.2. Participants

Participants were adult women living in the Busan and Gyeongnam regions who had been diagnosed with breast cancer. Specific inclusion criteria were as follows: over-middle-aged women, aged 40 years or older, who understood the study purpose and agreed to participate; those who were diagnosed with breast cancer and underwent mastectomy or partial mastectomy; and those who completed adjuvant treatments such as chemotherapy (excluding hormone therapy) and radiation therapy, or no longer required adjuvant treatment after surgery.

The number of participants was calculated using G*Power 3.1.9.4. For a hierarchical regression analysis, a minimum sample size of 129 was calculated with a significance level (α) of 0.05, effect size (f2) of 0.15, power (1 − β) of 0.80, and 10 predictors (general characteristics_age, religious state, and employment status; disease-related characteristics_treatment period, activity ability and symptomatic state; post-traumatic growth, successful aging, intolerance of uncertainty, and resilience) based on previous research [33,34]. One hundred and fifty people were surveyed, considering a 10% dropout rate. A total of 143 copies of the survey were analyzed, excluding 7 copies with insufficient responses.

### 2.3. Measures 

#### 2.3.1. Post-Traumatic Growth

Post-traumatic growth was measured using the Korean Version of the Post-traumatic Growth Inventory developed by Tedeschi and Calhoun [11], which was modified by Song Seung-hoon et al. [35]. This scale measured the extent to which one agrees with positive changes after a traumatic experience on a six-point scale ranging from 0 to 5 points, with higher scores indicating more positive changes after a traumatic experience. At the time of scale development, Cronbach’s α = 0.90. In this study, Cronbach’s α = 0.93, indicating excellent internal consistency.

#### 2.3.2. Successful Aging

Successful aging was measured using the Successful Aging Inventory developed by Troutman et al. [36], which was modified and supplemented by Jang Hyung-sook [37]. A five-point scale from 0 to 4 points was developed, with higher scores indicating higher levels of successful aging. At the time of scale development, Cronbach’s α = 0.86. In the study of Jang Hyung-sook [37], Cronbach’s α = 0.93. In this study, Cronbach’s α = 0.93, indicating excellent internal consistency.

#### 2.3.3. Intolerance of Uncertainty

Intolerance of uncertainty was measured using the Intolerance of Uncertainty Scale by Freeston et al. [24], which was adapted into Korean by Choi Hye-kyung [38]. The scale consists of 27 items scored on a four-point Likert scale, with higher scores indicating higher levels of intolerance of uncertainty. At the time of scale development, Cronbach’s α = 0.91. In this study, Cronbach’s α = 0.95, indicating excellent internal consistency.

#### 2.3.4. Resilience

Resilience was measured using the Connor–Davidson Resilience Scale 2, developed by Vaushnavi, Connor, and Davidson [39], after obtaining the Korean version from the developer with permission to use the scale. The scale consisted of two items scored on a five-point scale (0–4 points), with higher scores indicating higher levels of resilience. At the time of scale development, Cronbach’s α = 0.89. In this study, Cronbach’s α = 0.78, indicating acceptable internal consistency.

### 2.4. Data Collection

Data were collected from 21 January to 10 December 2021, after obtaining the approval of the Institutional Review Board of Kosin University Gospel Hospital (no: 2020-12-020). With the hospital’s permission, patients who met the inclusion criteria were referred by their physicians; the first and third authors personally explained the study purpose and sought patients’ consent to participate before or after the start of their medical care. Those who expressed their willingness to participate were given a consent form and asked to complete a questionnaire. The informed consent form specified the purpose, procedures, and methods, the method of participation and the capacity to withdraw at any time, and the protection of personal information. On average, it took 15 to 20 min to complete the questionnaire; if it was difficult for the participant to complete the questionnaire by themselves, the authors read the questionnaire to them and completed it for them. Completed questionnaires were sealed in a paper envelope to protect privacy and collected by the authors. In return for completing the survey, a gift coupon of 10,000 won (approximately 10 US dollars) was given to participants.

### 2.5. Data Analysis

Data were analyzed using SPSS 25.0 (IBM, Armonk, NY, USA) and PROCESS Macro version 3.5. General and disease-related characteristics, post-traumatic growth, successful aging, resilience, and intolerance of uncertainty were measured using descriptive statistics. Differences in post-traumatic growth, successful aging, resilience, and intolerance of uncertainty according to general characteristics were analyzed with *t*-tests and analyses of variance, with Scheffé’s post hoc tests. Correlations between post-traumatic growth, successful aging, resilience, and intolerance of uncertainty were analyzed using Pearson’s correlation tests. To testing the serial multiple mediation analysis, Process Macro model 6 and bootstrapping were uesed with a 95% confidence interval (CI).

## 3. Results

### 3.1. Participants’ General and Disease-Related Characteristics

Participants’ mean age was 52.47 years, with an average treatment duration of 34.5 months. Regarding their ability to perform activities at the time of the survey, 90 respondents (62.5%) were asymptomatic (able to perform activities as they did without the disease), and 54 respondents (37.5%) were symptomatic but fully mobile (e.g., light housework; Table 1).

### 3.2. Correlations of Post-Traumatic Growth, Successful Aging, Resilience, and Intolerance of Uncertainty in Participants

Successful aging in this study was positively correlated with post-traumatic growth (r = 0.708, *p* < 0.001) and resilience (r = 0.463, *p* < 0.001), and negatively correlated with intolerance of uncertainty (r = −0.282, *p* = 0.001). Post-traumatic growth was positively correlated with resilience (r = 0.318. *p* < 0.001), and resilience was negatively correlated with intolerance of uncertainty (r = −0.350, *p* < 0.001). (Table 2).

### 3.3. Mediating Effects of Resilience and Intolerance of Uncertainty on the Relationship between Post-Traumatic Growth and Successful Aging 

Multicollinearity was examined prior to analyzing the mediating effect of resilience and intolerance of uncertainty on the relationship between post-traumatic growth and successful aging of participants. Subsequently, the tolerance limits were between 0.806 and 0.897, which was above 0.10; the variance of the inflation factor was between 1.115 and 1.240, which was below 10, indicating no problem with multicollinearity. Further, the Durbin–Watson index was 2.180, which was close to 2; the P-P plot showed a normal distribution, and the scatterplot of the standardized residuals confirmed homoscedasticity. 

To test the mediating effect of resilience and intolerance of uncertainty on the relationship between post-traumatic growth and successful aging, religion, job, and activity ability, which showed significant differences in successful aging, were controlled and analyzed using the PROCESS macro model 6. Post-traumatic growth had a significant effect on the mediator resilience (B = 0.284, *p* < 0.001), and resilience had a significant effect on successful aging (B = 0.202, *p* < 0.001).

The direct effect for the successful aging of post-traumatic growth was 0.467 and was significant with a 95% bootstrap confidence interval (0.439 to 0.639) that did not include zero. The indirect effect of post-traumatic growth on successful aging, mediated by resilience, was 0.057, which was significant with a 95% bootstrap confidence interval (0.023 to 0.102) that did not include zero. The indirect effect of post-traumatic growth on successful aging, mediated by intolerance of uncertainty, was 0.006, which was not significant with a 95% bootstrap confidence interval (−0.009 to 0.027) that included zero. The indirect effect of post-traumatic growth on successful aging through both resilience and intolerance uncertainty was 0.009, which was also not significant with a 95% bootstrap confidence interval (−0.002 to 0.023) that included zero. (Table 3; Figure 1).

## 4. Discussion

This study examined the mediating effects of resilience and intolerance of uncertainty on the relationship between post-traumatic growth and successful aging in breast cancer survivors. Our study provided several noteworthy findings that contribute to the understanding of these psychological constructs and their interplay in the context of successful aging. 

One of the most salient findings is the mediating role of resilience in the relationship between post-traumatic growth and successful aging. Not only did post-traumatic growth significantly affect resilience, but resilience, significantly affected successful aging. This suggests that individuals who experience post-traumatic growth might foster a greater sense of resilience, which subsequently contributes to more successful aging.

Although direct comparisons of results were difficult owing to the lack of studies that comprehensively examined the relationship between these variables in patients with breast cancer, the current results supported previous research suggesting that post-traumatic growth [8,40] and resilience [41] influenced successful aging. An individual’s resilience to cancer is the personal trait that protects him or herself against the disease and positively influences recovery [42]. Increased resilience to cancer promotes successful adaptation to one’s condition. A previous study also reported that resilience to cancer was associated with positive mental health, quality of life, an improved pain threshold, increased physical activity, and improved rehabilitation outcomes in patients with cancer [43]. Promoting resilience focuses on interventions that address cognitive and problem-solving skills in stressful situations or focus on an individual’s self-esteem or positive emotions to help him or her overcome negative situations by improving these internal characteristics [44]. Therefore, it is necessary to develop and apply programs to promote the resilience of patients with breast cancer through education on how to promote resilience, manage stress, and strengthen various internal characteristics of breast cancer survivors. Further, the findings indicate that as resilience increases, intolerance to uncertainty decreases. This is similar to previous research [21] that found a negative correlation between intolerance of uncertainty and resilience. Individuals with higher resilience are more likely to be resilient when faced with uncertainty and, thus feel less stress or anxiety about uncertainty as compared to their counterparts. This can occur through both direct and indirect effects, suggesting that increasing resilience can help improve one’s ability to cope with uncertainty. Resilience could affect uncertainty tolerance through some mediating variables (e.g., self-efficacy and positive thinking). Hence, future research should explore which variables play this mediating role.

In contrast to resilience, intolerance of uncertainty did not play a significant mediating role in the relationship between post-traumatic growth and successful aging. This suggests that while intolerance of uncertainty could be relevant in other contexts, its role in the trajectory from post-traumatic growth to successful aging might be minimal or overshadowed by other more dominant factors, such as resilience. It was difficult to compare results with those of other studies, as research confirming the mediating effect of intolerance of uncertainty in the relationship between post-traumatic growth and successful aging was not available. However, this differed from the significant positive association between post-traumatic growth and intolerance of uncertainty in Jewish pregnant women [45] and from uncertainty acting as a significant barrier to successful aging in midlife [46]. Intolerance of uncertainty in breast cancer survivors could have had a lack of mediating effect on the relationship between post-traumatic growth and successful aging owing to a combination of factors, including individual characteristics and socio-psychological factors. Tolerance of uncertainty is a personality trait; people who have it can manage situations well in uncertainty without experiencing destructive anxiety [47]. Therefore, intolerance of uncertainty in breast cancer survivors could also affect many of the situations they experience. Further, since both post-traumatic growth and successful aging were negatively correlated with intolerance of uncertainty, future research should further examine the role of intolerance of uncertainty in the relationship between post-traumatic growth and successful aging. Moreover, the role that negative variables, including depression and anxiety, along with intolerance of uncertainty, play in the relationship between post-traumatic growth and successful aging should be further explored. 

Beyond the mediators, it is essential to acknowledge the direct impact of post-traumatic growth on successful aging. Our results indicate that even after accounting for the roles of resilience and intolerance of uncertainty, post-traumatic growth has a strong effect on successful aging. This was consistent with previous studies on post-traumatic growth that positively influenced successful aging in breast cancer survivors and middle-aged women [8,9]. Previous research has identified personal characteristics, disease factors, cognitive processes, coping strategies, social support, religion, spirituality, body roles, and physical activity as influencing post-traumatic growth in women with breast cancer [48]. This means that it is necessary to fully understand the post-traumatic growth of breast cancer survivors; develop and implement programs to support their post-traumatic growth and consequently help them age successfully; and consider the many variables that affect post-traumatic growth when developing programs to support post-traumatic growth. Future research will, therefore, need to identify which elements of post-traumatic growth are more important for successful aging.

Among the general characteristics and disease- and treatment-related characteristics of participants, the characteristics that made a significant difference in successful aging were “religion,” “job,” and “activity ability.” Kim - and Kim [49] found significant differences in education level and job but not in the stage of cancer, time after surgery, type of surgery, chemotherapy, radiotherapy, targeted therapy, hormone therapy, or self-help groups for all disease-related characteristics among middle-aged breast cancer survivors. This was similar to previous research [8] that suggested that the loss of physical function or frailty in daily life did not have a direct effect on successful aging in breast cancer survivors. Coping processes, such as post-traumatic growth, contributed to successful aging rather than health status. Nevertheless, physical issues are important in aging, and biological factors and diseases have a significant impact on successful aging [50]. Therefore, in the future, it is necessary to confirm the relationship between each variable and successful aging in repeated studies with a larger sample size and patients with different types of cancer.

### Limitations and future research Suggestions

This study was limited in the demographic variables that were collected, as well the ability to determine the response rate during the data collection phase. Data collection took place over a relatively short period. Participation in the survey was voluntary, which could, have resulted in unintended bias. This study only collected data from respondents from the Busan and Gyeongnam regions, and participants were outpatients at a single hospital. This could impact the generalizability of the results compared to the broader South Korean population. In addition, it did not reflect the relationship between the main variables and a wider range of variables (e.g., well-being, quality of life, depression, etc.), which should be considered in the future. This study can be used as a basis for identifying the extent of post-traumatic growth in breast cancer survivors and exploring ways to strengthen their resilience while developing nursing intervention programs to promote the successful aging of breast cancer survivors in clinical settings.

## 5. Conclusions

Successful aging among breast cancer survivors was positively correlated with post-traumatic growth and resilience and negatively correlated with intolerance of uncertainty. A mediating effect of resilience on the relationship between post-traumatic growth and successful aging was identified. There is a need to develop and apply intervention programs that consider post-traumatic growth and enhance resilience for successful aging in breast cancer survivors.

## Figures and Tables

**Figure 1 healthcare-11-02843-f001:**
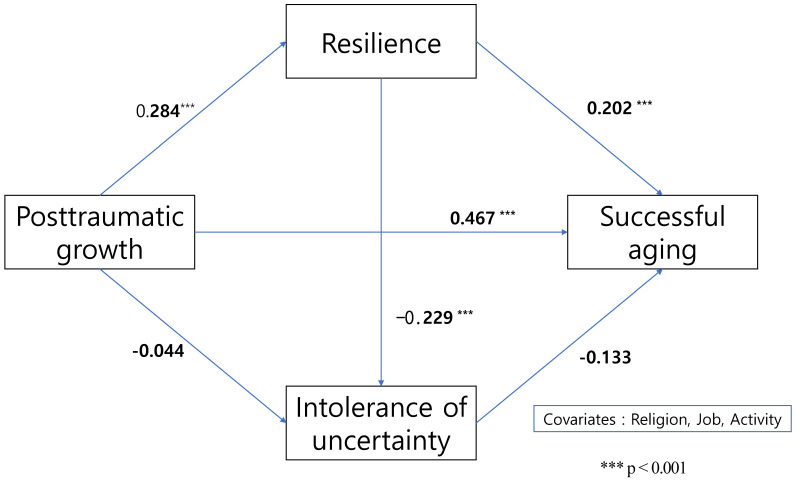
Mediating effects of resilience and intolerance of uncertainty on post-traumatic growth in successful aging.

**Table 1 healthcare-11-02843-t001:** Characteristics of participants (N = 143).

Characteristic	Category	Mean (±SD) or n (%)
Age (years)		52.47 (±8.23)
Treatment period (months)		34.46 (±25.95)
Religious	Yes	80 (55.9%)
	No	63 (44.1%)
Employed	Yes	68 (47.6%)
	No	75 (52.4%)
Activity ability	Asymptomatic	89 (62.2%)
	Symptomatic but fully functional	54 (37.8%)

**Table 2 healthcare-11-02843-t002:** Correlations among variables (N = 143).

	Successful Aging	Post-Traumatic Growth	Resilience	Intolerance of Uncertainty
Post-traumatic growth	0.708 ***			
Resilience	0.463 ***	0.318 ***		
Intolerance of uncertainty	−0.282 ***	−0.155 (0.063)	−0.350 ***	1
Mean	2.71	3.38	2.99	2.31
SD	0.71	0.90	0.77	0.55

*** *p* < 0.001.

**Table 3 healthcare-11-02843-t003:** Mediating effects of resilience and intolerance of uncertainty on post-traumatic growth and successful aging (N = 143).

	Unstandardized Coeff.	SE	95% CI (Lower)	95% CI (Upper)	Standardized Effect	*p*
Total effect	0.539	0.051	0.439	0.639	0.762	<0.001
Direct effect	0.467	0.050	0.368	0.566	0.661	<0.001
x→ m1 → y	0.057	0.020	0.023	0.102	0.073	
x → m2 → y	0.006	0.009	−0.009	0.027	0.007	
x → m1 → m2 → y	0.009	0.007	−0.002	0.023	0.011	
Total indirect effect	0.072	0.024	0.029	0.122	0.092	

Note: Number of bootstrap samples for bias-corrected bootstrap confidence intervals; 5000. Level of confidence for all confidence intervals; 95%. x = post-traumatic growth; m1 = resilience, m2 = intolerance of uncertainty, y = successful aging.

## Data Availability

The data presented in this study are available on request from the corresponding author. The data are not publicly available owing to privacy restrictions.

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
