# Peer review of "Effects of Post Traumatic Growth on Successful Aging in Breast Cancer Survivors in South Korea: The Mediating Effect of Resilience and Intolerance of Uncertainty"

_healthcare, 2023, doi:10.3390/healthcare11212843_

Round 1
Reviewer 1 Report
Comments and Suggestions for Authors
The structure of the literature review, inference, and discussion in this study needs further improvement. Here are some suggestions:
1. The link between post-traumatic growth and successful aging appears to be relatively tenuous, with only one relevant paper found among the available literature [7]. It is advisable to enhance the association between these two factors.
2.Could you clarify the connection between resilience and intolerance of uncertainty? The existing information does not provide sufficient clarity for me to comprehend this relationship.
3. Discussing the content, including future research, it is suggested to add a section describing future research and research limitations. (Line261-270, Line 284-289, 301-303, 307-310)
4.Line 259-261, Please conclude the study on the relationship between post-traumatic growth and successful aging."
5.Please add a section discussing research limitations
6.This study did not discover a significant association between intolerance of uncertainty and successful aging, while the significance of resilience has been consistently established in various prior research. Consequently, the research findings lack novelty. Are there any distinct facets or noteworthy elements in this study?
Author Response
We are happy to resubmit a revised version of manuscript ID healthcare-2623860, “Effects of Post-traumatic Growth on Successful Aging in Breast Cancer Survivors in South Korea: The Mediating Effect of Resilience and Intolerance of Uncertainty” for publication in Healthcare. We appreciate the opportunity to revise and resubmit this manuscript. We have read the comments of the reviewers and have revised the manuscript accordingly; we believe our manuscript has benefited immensely from these insightful suggestions for revision. We look forward to working with you and the reviewers to move this manuscript closer to publication in Healthcare.
All comments by the reviewers and editor have been addressed, with corresponding changes made directly to the manuscript where appropriate. We have numbered the changes to make them easier to identify.
Thank you for your consideration. I look forward to hearing from you.

Reviewer 2 Report
Comments and Suggestions for Authors
Please see attached

Comments on the Quality of English LanguageThere are minor editing issues
Author Response

(The authors gave the same response as above.)

Reviewer 3 Report
Comments and Suggestions for Authors
I appreciate the opportunity to review this article.
Summary: The authors do not address the number of participants and the type of study. This is also not evident in the title. The results are very general, you can develop this aspect further.
Introduction: authors must present the research question.
Provide adequate justification for the study.
Methodology: the authors present the study design in a clear way.
Results and discussion: support the results with bibliography relevant to the research, which facilitates understanding of the results obtained. They point out the limitations of the study appropriately.
Author Response

(The authors gave the same response as above.)

Reviewer 4 Report
Comments and Suggestions for Authors
great paper
only minimum details to add
see attached manuscript
you certainly provided the information to make it clear for the reader in particular in defining the main concepts

Author Response

(The authors gave the same response as above.)

Round 2
Reviewer 1 Report
Comments and Suggestions for Authors
The article was subjected to a thorough revision by the author to elevate its overall value. In closing, there's a modest recommendation: "Considering research constraints is advised, as the temporal separation (longitudinal design) between assessments of presumed cause and effect variables is presumed to enhance the confidence in establishing causal conclusions."